# Icariin Treatment Enhanced the Skeletal Response to Exercise in Estrogen-Deficient Rats

**DOI:** 10.3390/ijerph16193779

**Published:** 2019-10-08

**Authors:** Renqing Zhao, Wenqian Bu, Yingfeng Chen

**Affiliations:** 1College of Physical Education, Yangzhou University, 88 Daxue South Rd, Yangzhou 225009, Jiangsu, China; 2College of Physical Education and Health Sciences, Zhejiang Normal University, 688 Yingbin Avenue, Jinhua 321004, Zhejiang, China

**Keywords:** exercise, icariin, osteoporosis, estrogen, osteoblast

## Abstract

Estrogen deficiency frequently leads to a fall in estrogen receptor-α (ERα) numbers and then reduces the skeletal response to mechanical strain. It, however, is still unclear whether phytoestrogen administration will enhance the effects of exercise on the estrogen-deficient bone loss. This study aimed to determine the effect of Icariin treatment on the response of osteogenic formation to exercise in ovariectomized (OVX) rats. Thirty-two 3-month old female Sprague–Dawley rats were randomly allocated into four groups: (1) Sham-operated (SO); (2) OVX; (3) OVX plus exercise (EX); and (4) OVX plus exercise and Icariin (EI). After 8-week interventions, the rats were killed and samples were collected for bone morphometry, reverse transcription-polymerase chain reaction (RT-PCR), and Western blot analyses. EI interventions showed a greater improvement for the OVX-induced bone loss and the elevated serum tartrate-resistant acid phosphatase (TRAP) and alkaline phosphatase (ALP) compared with EX only. Both EX and EI interventions bettered the OVX-related reduction of BV/TV and trabecular number and thickness, and decreased the enlargement of trabecular bone separation (Tb. Sp); the improvement for BV/TV and Tb. Sp was greater in EI group. Furthermore, EX and EI treatment significantly increased the number of ALP^+^ cells and mineralized nodule areas compared with OVX group; the change was higher in EI group. Additionally, in comparison to OVX rats, the protein and mRNA expression of β-catenin, phosphorylated-Akt (p-Akt) or Akt, ERα, and Runt-related transcription factor 2 (Runx2) in osteoblasts were elevated in EX and EI intervention rats, with greater change observed in EI group. The upregulated β-catenin and Akt mRNA levels in EX and EI groups was depressed by ICI182780 treatment, and the difference in β-catenin and Akt mRNA levels between EX and EI groups was no longer significant. Conclusively, the combination of Icariin and exercise significantly prevent OVX-induced bone loss and increase osteoblast differentiation and the ability of mineralization compared with exercise alone; the changes might be regulated partly by ERα/Akt/β-catenin pathway.

## 1. Introduction

Osteoporosis is a common aged-related skeletal disorder characterized by compromised bone strength predisposing the older adults to high risk of fracture [1,2]. Osteoporosis-related medical care causes a heavy burden both on society and families [3,4]. To find effective therapeutic strategies becomes one emergent clinical research task [5,6]. Compelling evidence has confirmed that exercise generates beneficial effects on loading sites in osteoporotic individuals [7,8,9]; it has been recognized as an effective way to prevent age-related bone loss and a promising strategy for fracture reduction [10,11,12,13,14]. However, the exercise-associated benefits for bone mass are frequently weakened in some situations, for example, women with menopause.

Skeletal system continuously adapts its mineral materials and microstructure to daily mechanical loading to maintain a fine balance between bone absorption and formation. This physiological adaption is mainly controlled by estrogen receptor-α (ERα) signaling pathways [15]. Decreased estrogen concentrations in postmenopausal women frequently down-regulate the number and function of ERα and subsequently impair skeletal response to mechanical loading [16,17]. Therefore, upregulation of ERα expression is expected to increase the response of osteogenic formation to mechanical strain; this approach might improve the effect of exercise on the prevention of bone loss in postmenopausal women. However, the question has not been addressed clearly yet. Additionally, traditional hormone replacement therapy (HRT) is regarded to have beneficial effects on the number and function of ERα [16], but it is also suspected to link with severe side effects [18,19]. Therefore, alternative pharmacological therapy that can generate estrogen-like activity but has no the adverse effects seen in HRT is urgently needed.

Icariin, a maker flavonoid glycoside, is extracted from *Herba Epimedii*, which is used to treat skeletal diseases in traditional Chinese medicine for thousands of years [20,21]. Recent evidence has demonstrated that Icariin has the potential of estrogenic effects and shows improvement for bone loss in ovariectomized (OVX) mice [22,23]. Several studies suggested Icariin inhibited bone loss mainly by increasing osteoblastic differentiation [24,25,26]. Previous study [27] reported that the combination of Soy Isoflavone and exercise resulted in more BMD gains than exercise alone, but it remains unclear whether Icariin would upregulate skeletal response to exercise and the regulation mechanism is still unknown. Lau and colleagues [28] reported that mechanical strain increased bone formation mainly by upregulating ERα expression and then promoting Akt/β-catenin signaling pathways, which subsequently enhanced osteoblastic proliferation and differentiation. We hypothesized that Icariin treatment might potentially increase ERα expression and then promotes Akt/β-catenin signaling pathways, all of which subsequently enhance osteogenic formation in response to mechanical strain. Given the clinical importance of this issue, to elucidate it will provide a novel and effective approach for preventing postmenopausal bone loss. Therefore, our study aimed to examine the effects of the combination of Icariin and exercise on bone loss and the capacity of osteogenic formation in OVX rat. The changes in the expression of ERα, Akt, β-catenin, and Runt-related transcription factor 2 (Runx2) in osteoblasts were also determined.

## 2. Materials and Methods

### 2.1. Animals and Intervention Protocols

Thirty-two 3-month old female Sprague–Dawley (SD) rats (purchased from Kaixue Bio-Technique Co., Ltd., Shanghai, China) were housed under the temperature of 23 ± 2 ℃ and with a 12-h light-dark cycle. Food and drinking water were supplied *ad libitum*. One week after arrival, the rats were sham-operated (SO) or OVX according to experimental protocols, and randomly assigned to 4 groups in parallel (8 rats in each group): (1) SO group; (2) OVX group; (3) exercise (EX) group: OVX rats with exercise intervention for 8 weeks; (4) exercise and Icariin (El) group: OVX rats received exercise and Icariin interventions for 8 weeks. The experimental protocols were reviewed and approved by the Ethics Committee of Zhejiang Normal University (ethical code number KYZKYY14483).

Two weeks after operation, exercise rats received one week of adaptive training with a protocol of daily 20-min treadmill running and the speed gradually increasing from 12 meters/min to 16 meters/min (0% grade). After adaptive training, exercise rats were trained regularly for five days per week, with each training section about 60 min at a speed of 18 meters/min and a grade of 5%. After 3 weeks of operation, EI rats were also fed daily with 50 mg/kg Icariin by gavage (purchased from GuideChem co., Ltd., Hangzhou, China) besides regular exercise training. At the end of interventions, all rats were killed within 24 h. Blood samples were collected after anesthetized (10% chloral hydrate, 3 mL/kg body weight). Serum was separated by centrifugation at 1500g for 20 min at 4 °C and then kept at −20 ℃. Femora, tibia, and lumbar vertebra were dissected and stored in a freezer at −80 °C until analysis.

### 2.2. Bone Density and Morphometry Analyses

The fifth lumbar vertebrae were used for bone mineral density (BMD) analysis. BMD was measured by dual energy X-ray absorptiometry (DEXA) (Lunar Prodigy, GE Inc., Madison, WI, USA) using the manufacturer provided high-resolution software for small animals. The left femur was fixed in 4% paraformaldehyde and embedded in methyl methacrylate plastic after serial dehydration with a graded ethanol series to xylene. Five-μm sections were made, and HE staining was carried out by staining with haematoxylin for 3 min followed by 2 min of staining with eosin. Morphometry was determined by measuring the bone volume (BV/TV), trabecular thickness (Tb. Th), trabecular number (Tb. N), trabecular spacing (Tb. Sp), and analyzed by OsteoMeasure software (Osteometrics, Atlanta, GA, USA) under electron microscope (DM400, Leica, Solms, German).

### 2.3. Bone Mesenchymal Stem Cell Culture

Bone mesenchymal stem cells (BMSCs) were collected as previously described [29]. Briefly, under aseptic conditions, BMSCs were flushed from bone marrow with α-minimal essential medium (α-MEM) (Invitrogen, Carlsbad, CA, USA). Then the cells were re-suspended and cell culture medium was replaced every 3 days. For osteogenic differentiation, the culture medium was replaced with osteogenic medium (α-MEM supplemented with 15% fetal calf serum plus 1% penicillin/streptomycin, 100 nM dexamethasone, 50 μg/mL ascorbate-2-phosphate and 10 mM β-glycerol phosphate). The medium was changed every 3 days. For ERα blocking test, the culture system on day 7 was treated with either placebo or 200 nM ICI182780 (GlpBio, co., Ltd., Montclair, NJ, USA) to block ERα pathways.

### 2.4. Alkaline Phosphatase Staining and Alizarin Red Staining

At day 7 of osteogenic differentiation culture, the cells were washed in phosphate-buffered saline (PBS), fixed in 10% paraformaldehyde for 10 min at room temperature, and rinsed in distilled water. The alkaline phosphatase (ALP) staining mixture was added for 30 min at room temperature in the dark. The cells were rinsed in distilled water and PBS to reduce non-specific staining. At day 14 of culture, cells were washed with phosphate buffered saline, fixed with 70% ethanol for 1 h, washed 3 times with distilled water, and stained with 40 nM Alizarin red (Sigma-Aldrich Corp., St. Louis, MO, USA) for 10 min. Then the staining of calcium mineral deposits was quantified.

### 2.5. Serum E_2_, ALP and Tartrate-Resistant Acid Phosphatase Analyses

The serum concentrations of E_2_, ALP and tartrate-resistant acid phosphatase (TRAP) were determined with ELISA kits (R&D Systems, Inc., Minneapolis, MN, USA), according to the instructions in the manufacturer’s protocol.

### 2.6. Western Blot Analysis

On the 7th day of osteogenic differentiation culture, cells were lysed using the RIPA lysis buffer containing 50 mM Tris–HCl, pH 8.0, 150 mM NaCl, 1% NP-40, 0.5% sodium chloride, 0.1% sodium dodecyl sulfate (Sigma-Aldrich Corp., St. Louis, MO, USA). Extracts were fractionated by SDS-PAGE and transferred to a Trans-Blot Nitrocellulose membrane (BioRad, Hercules, CA, USA). After blocking with 5% nonfat dry milk in Tris-buffered saline (TBS), we incubated the membranes overnight at 4 °C with antibody to β-catenin, ERα, p-Akt, or Runx2. For loading control, we used antibodies to β-actin. The secondary antibody was diluted to 1:1000 and incubated with the membrane for 2 h at room temperature. After the last washing step, 5-Bromo-4-chloro-3-indolyl phosphate-nitro blue tetrazolium (NBT-BCIP) (Zymed, Laboratory Inc., San Francisco, CA, USA) detection was carried out following the manufacturer’s instructions.

### 2.7. RT-PCR Analysis

After 7 days of osteogenic differentiation culture, cells were collected to extract total RNA using Trizol reagent (Gibco-BRL, Rockville, MD, USA) and isolated as specified by the manufacturer. The RNA was DNAse-treated (DNase I-RNase-Free, Ambion) to remove any contaminating DNA; 200 ng of total RNA was reverse-transcribed with oligo dT primers using the High Capacity cDNA RT Kit (Applied Biosystems, Foster City, CA, USA) in a 20-μL cDNA reaction, as specified by the manufacturer. For quantitative PCR, the template cDNA was added to a 20 μl reaction with SYBR Green PCR Master Mix (Applied Biosystems, Foster City, CA, USA) and 0.2 μM of primer. The amplification was carried out using an ABI Prism 7000 for 40 cycles under the following conditions: an initial denaturation of 95 °C for 10 min, plus 40 cycles of 95 °C for 15s, then 60 °C for 1 min. The fold changes were calculated relative to β-actin using the ΔΔ Ct method for Akt, ERα, β-catenin, and Runx2 mRNA analysis. The following primer sets were used: Akt: forward, 5′-GCAGCACGTGTACGAGAAGA-3′; reverse, 5′-GGTGTCAGTCTCCGACGTG′; ERα: forward, 5′-GCCATCAAGAAGATCAGCC-3′; reverse, 5′-CGTAGCCACATACTCCGTCA-3′; β-catenin: forward, 5′-TCAGGAAAGCAAGCTCATCATTC-3′; reverse, 5′-ACGATGGCCGGCTTGTT-3′; Runx2, forward, 5′-GCCGGGAATGATGAGAACTA-3′; reverse, 5′-GGTGAAACTCTTGCCTCGTC-3′; β-actin: forward, 5′-GTACGCCAACACAGTGCTG-3′; reverse, 5′-CGTCATACTCCTGCTTGCTG-3′.

### 2.8. StatisticalAnalysis

Statistical analysis was performed using STATA software (Version 15, StataCorp LP, College Station, TX, USA). Statistical significance in weight, BMD, serum estrogen, ALP and TRAP, and the percentage of protein and gene expression was determined by analysis of variance (ANOVA); differences between means were evaluated using Student’s *t*-test. A level of *p* < 0.05 was accepted as significant.

## 3. Results

### 3.1. General Characteristics

Table 1 showed that the baseline weight of rats was not significantly different between the four groups. After 8-week intervention, greater weight gains were found in OVX, EX, and EI groups, with the largest increment in OVX group. OVX induced a significant bone loss, but EX and EI interventions markedly alleviated bone wasting, with more BMD increment found in EI group. OVX potently elevated the serum levels of ALP and TRAP and decreased E_2_ concentrations. Both EX and EI interventions decreased serum biomarkers and elevated E_2_ concentrations, and the beneficial changes in E_2_, ALP, and TRAP were greater for EI treatment compared with EX intervention only (Table 1).

### 3.2. Bone Tissue Characteristic Analysis

OVX significantly decreased BV/TV, Tb.N and Tb.Th, and increased Tb. Sp (Table 2). Both EX and EI interventions improved these adverse changes; EI generated greater BV/TV gains and decreased more Tb.Sp compared with EX only (Table 2).

### 3.3. Osteogenic Differentiation and Deposition

OVX induced a reduction of ALP^+^ cell numbers and mineralized nodule areas compared with the SO group (Figure 1). EX and EI interventions significantly promoted the osteoblastic differentiation and osteogenic deposition; the beneficial changes were greater for EI interventions than EX alone (Figure 1).

### 3.4. Western Blot and PCR Analyses

Both protein and mRNA expression of ERα, p-Akt (Akt) or Akt, β-catenin, and Runx2 were reduced by OVX, and EX and EI interventions significantly elevated the decreased levels. The protein and mRNA levels of ERα, β-catenin, and p-Akt (Akt) or Akt were higher in EI than EX groups (Figure 2 and Figure 3). To examine whether Icariin enhanced the skeletal response to exercise through promoting ERα expression, we pre-treated the osteoblastic cultures with ICI182780. The EX- and EI- induced increment of β-catenin and Akt mRNA levels was significantly decreased by ICI182780 treatment (Figure 4), and β-catenin and Akt mRNA levels were no longer different between EI and EX groups unlike that treated with placebo (Figure 4).

## 4. Discussion

Our study found that both exercise and exercise combined with Icariin were effective in preventing estrogen-deficient bone loss, and significantly promoted osteoblast formation and the ability of mineralization in OVX rats. Greater changes were frequently found in the combination of exercise with Icariin. Icariin elevated the skeletal response to exercise partly through upregulating ERα expression because higher mRNA expression of β-catenin and Akt found in EI interventions were attenuated by anti-ERα agents.

Exercise for the prevention of osteoporosis has been explored previously [30]. But questions remained about the regulating mechanism of this favorable effects. Our findings indicated that exercise prevented bone loss mainly by promoting osteogenic formation and mineralization, and ERα/Akt/β-catenin signaling pathway might partly contribute to this benefit change. Previous ex vivo evidence demonstrated that mechanical strain induced osteogenic generation mainly by increasing ERα expression in osteoblasts [31]. Upregulated ERα expression helps β-catenin translocate into nuclei where β-catenin is combined with T cell factor (TCF) and lymphoid enhancer factor (LEF) to activate TCF/LEF responsive target genes [31]. The ERα-induced activation and increment of β-catenin is partly regulated by Akt phosphorylation [28]. Our study provided in vivo evidence to ascertain this regulation pathway in exercise-intervention OVX rats because both phosphorylated-Akt (p-Akt) and β-catenin were elevated by exercise and regulated by anti-ERα agent.

Use of Chinese herbs to improve skeletal response to exercise is a novel approach to the treatment of osteoporosis. Eight-week interventions of Icariin and exercise resulted in greater improvement of osteoporotic status compared with exercise alone. Wu et al. [27] reported that combined interventions of food supplementations (Soy Isoflavone) and exercise resulted in more BMD gains than exercise alone. However, the regulation mechanism was not clear. Our study suggested that more BMD gains found in Icariin and exercise interventions were linked with higher capacity of osteoblastic differentiation and mineralization. Furthermore, the beneficial changes might partly be regulated by elevated ERα expression. Icariin is a potential agent for increasing ERα signalings [24], in agreement with which our results found that the higher ERα expression was seen in the combined interventions of irisin and exercise than exercise only. As reported in previous study [28], ERα plays an essential role in modulating the effects of mechanical strain on bone formation. As a result, the increment of ERα subsequently caused higher levels of Akt and β-catenin, which then augment the proliferation and differentiation of osteoblasts. Interestingly, this regulating axis was affected by ICI18278. When treated with ICI182780, the mRNA levels of β-catenin and Akt significantly decreased and the difference between EI and EX groups was not obvious. It is suggested that Icariin elevated skeletal response to exercise partly thorough upregulating ERα expression and then triggering Akt/β-catenin pathways.

Recently, Chinese herbs showed advantages in counteracting some diseases. A successful case was seen in the treatment of Plasmodium falciparum malaria in which Artemisinin-based combination therapies are the first-line therapy strategy [32]. Given the obvious adverse effects of HRT, Icariin might be a promising alternative therapy strategy for the prevention of osteoporosis in postmenopausal women.

## 5. Conclusions

Our study has determined the combined benefits of exercise and Icariin for improving OVX-induced bone loss by up-regulating osteoblastic formation, but there remain some questions undetermined, for example, whether the combined interventions would affect osteoclast differentiation is still unclear. To elucidate this question will better our understandings about the mechanism regulating the effects of Icariin on skeletal response to exercise.

## Figures and Tables

**Figure 1 ijerph-16-03779-f001:**
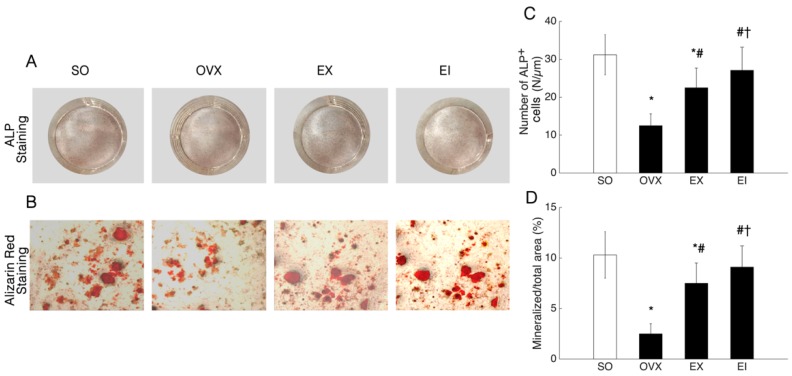
ALP^+^ cell number and mineralized nodule area. SO: Sham-operated; OVX: ovariectomized; EX: exercise; EI: exercise and Icariin. 1. Comparison of the groups of OVX, EX, and EI with sham group: * <0.05. 2. Comparison of EX and EI groups with OVX group: # <0.05. 3. Comparison between EX with EI group: † <0.05.

**Figure 2 ijerph-16-03779-f002:**
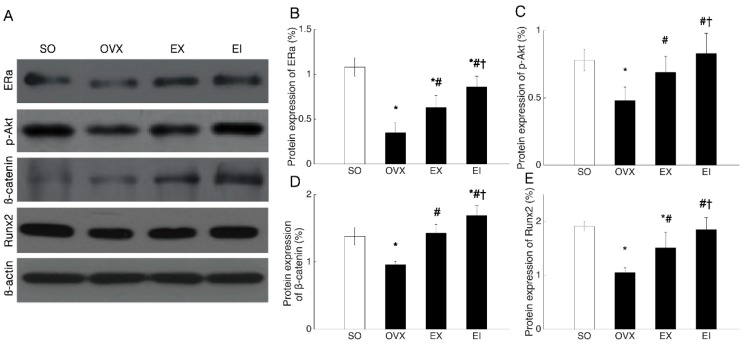
The protein expression of ERα, p-Akt, β-catenin, and Runx2 in osteoblast culture. SO: Sham-operated; OVX: ovariectomized; EX: exercise; EI: exercise and Icariin. 1. Comparison of the groups of OVX, EX, and EI with SO group: * <0.05. 2. Comparison of EX and EI groups with OVX group: # <0.05. 3. Comparison between EX with EI group: † <0.05.

**Figure 3 ijerph-16-03779-f003:**
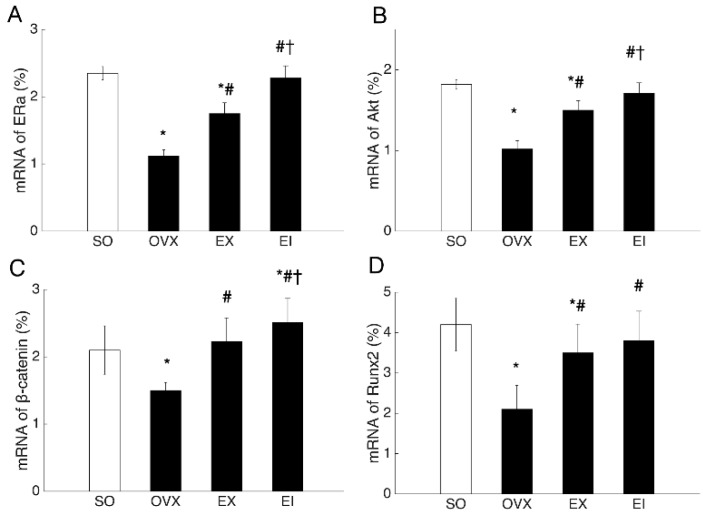
The percentage of ERα, Akt, β-catenin, and Runx2 mRNA in osteoblast culture. SO: Sham-operated; OVX: ovariectomized; EX: exercise; EI: exercise and Icariin. 1. Comparison of the groups of OVX, EX, and EI with SO group: * <0.05. 2. Comparison of EX and EI groups with OVX group: # <0.05. 3. Comparison between EX with EI group: † <0.05.

**Figure 4 ijerph-16-03779-f004:**
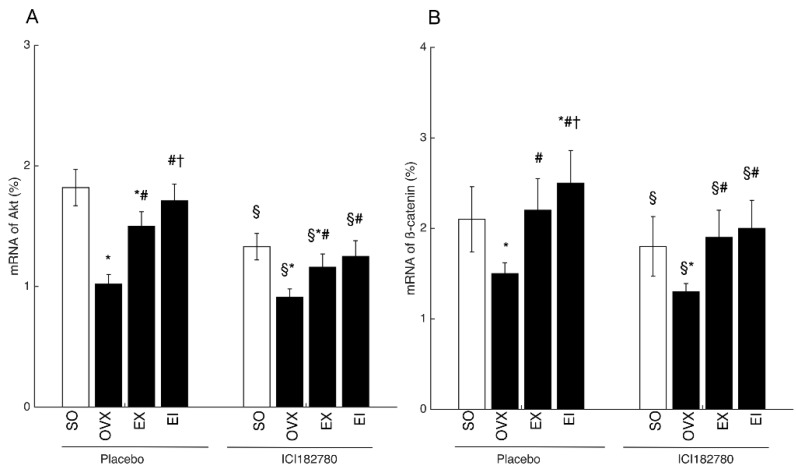
The effects of ICI182780 treatment on the mRNA expression of Akt and β-catenin. SO: Sham-operated; OVX: ovariectomized; EX: exercise; EI: exercise and Icariin. 1. Comparison of the groups of OVX, EX, and EI with SO group: * <0.05. 2. Comparison of EX and EI groups with OVX group: # <0.05. 3. Comparison between EX with EI group: † <0.05. 4. Comparison between placebo and ICI182780 treatment: § <0.05.

**Table 1 ijerph-16-03779-t001:** Physical and serum parameters of rats between different treatment groups.

Variables	SO	OVX	EX	EI
Weight _pre_ (g)	210.70 ± 14.95	212.46 ± 14.60	211.01 ± 9.21	209.25 ± 14.69
Weight _post_ (g)	288.36 ± 15.51	323.65 ± 16.09 *	307.48 ± 12.94 *^#^	302.67 ± 13.72 *^#^
BMD _spine_ (g/cm^2^)	0.175 ± 0.01	0.155 ± 0.011 *	0.162 ± 0.01 *^#^	0.171 ± 0.01 ^#†^
Serum E_2_ (pg/mL)	23.61 ± 2.27	12.94 ± 2.95 *	17.33 ± 1.81 *^#^	20.50 ± 1.98 *^#†^
Serum ALP (IU/dl)	9.67 ± 2.82	15.69 ± 3.69 *	12.52 ± 2.75 *^#^	9.39 ± 2.12 ^#†^
Serum TRAP (IU/dl)	39.23 ± 6.21	69.12 ± 8.61 *	51.25 ± 7.83 *^#^	40.53 ± 6.39 ^#†^

Note: SO: Sham-operated; OVX: ovariectomized; EX: exercise; EI: exercise and Icariin; BMD: bone mineral density; ALP: alkaline phosphatase; TRAP: Tartrate-resistant acid phosphatase; 1. Comparison of the groups of OVX, EX, and EI with SO group: * <0.05. 2. Comparison of EX and EI groups with OVX group: ^#^ <0.05. 3. Comparison between EX with EI group: ^†^ <0.05.

**Table 2 ijerph-16-03779-t002:** Changes in morphological structure of bone.

Variables	SO	OVX	EX	EI
BV/TV (%)	59.2 ± 8.5	32.6 ± 6.3 *	45.2 ± 7.2 *^#^	55.5 ± 7.6 ^#†^
Tb.Th (mm)	83.1 ± 13.5	63.5 ± 7.3 *	78.3 ± 12.1 ^#^	80.9 ± 11.1 ^#^
Tb.N (N/mm^2^)	13.2 ± 2.1	7.3 ± 1.9 *	10.1 ± 2.5 *^#^	12.8 ± 2.9 ^#^
Tb.Sp (mm)	103.3 ± 19.5	149.2 ± 27.3 *	128.5 ± 21.8 ^#^	106.8 ± 23.6 ^#†^

**Note:** SO: Sham-operated; OVX: ovariectomized; EX: exercise; EI: exercise and Icariin; BV/TV: trabecular bone volume; Tb.N: trabecular number; Tb.Th: trabecular thickness; Tb.Sp: trabecular separation; 1. Comparison of the groups of OVX, EX, and EI with SO group: * <0.05. 2. Comparison of EX and EI groups with OVX group: ^#^ <0.05. 3. Comparison between EX with EI group: ^†^ <0.05.

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
