# Peer review of "Icariin Treatment Enhanced the Skeletal Response to Exercise in Estrogen-Deficient Rats"

_ijerph, 2019, doi:10.3390/ijerph16193779_

Round 1

Reviewer 1 Report

Manuscript entitled “Icariin treatment enhanced the skeletal response to exercise in estrogen-deficient rats” by Zhao et al shows the beneficial effects of Icariin on the skeletal system in vivo potentially through an ERα-mediated pathway. The manuscript explores a novel concept to that may eventually provide therapeutic benefit to alleviate osteoporosis, a detailed critique is as follows-

Overall, the manuscript needs a thorough rewrite with proper English formatting. The abstract defines methodology, which is not necessary. Instead, authors can explain why the study was done and define the first time usage of abbreviations such as TRAP. Methods are sufficient except in each subheading, authors should clearly state what cells on what days of culture were used for experiment. A flow chart of the time regime of in vivo studies will be helpful. Authors mention in Materials and Methods 2.1 line 64 “rats were fed regularly and allowed free access to water”, what does regularly mean- free access to food or timed? In the end of section 2.1 in Methods, describe how blood was collected and which other tissues were harvested and then again mention in the subheadings what tissues or cells were utilized for that particular method. Introduction and Discussion need to be expanded, authors should explain why the study was done, what are the existing studies and more importantly, how the mechanisms as to why it is important to study p-Akt and β-Catenin, importance of staining with Alizarin etc.?

Reviewer 2 Report

The article is written well, the ideas are clear, and the experimental data can basically explain its conclusions. But there are still a few points to discuss.

The quality of the image in this article is not clear, please provide high-definition pictures. Why antagonists choose ICI182780 instead of MPP? MPP is a specific antagonist of ERα, but ICI can antagonize ERα and ERβ. What is the dose-effect relationship of icariin? Has the author done a toxicity test? What is the effect of icariin and epimedium? If the effect of icariin is not obscene, I think this study is meaningless.

Round 2

Reviewer 2 Report

The article is written well, the ideas are clear, and the experimental data can basically explain its conclusions. But there are still a few points to discuss.

The quality of the image in this article is not clear, please provide high-definition pictures. Why antagonists choose ICI182780 instead of MPP? MPP is a specific antagonist of ERα, but ICI can antagonize ERα and ERβ. What is the dose-effect relationship of icariin? Has the author done a toxicity test?